## REVIEW ARTICLE

# Exploring the evidence for epigenetic regulation of environmental influences on child health across generations

Carrie V. Breton [1✉], Remy Landon[1], Linda G. Kahn [2],
Michelle Bosquet Enlow[3], Alicia K. Peterson[1], Theresa Bastain[1], Joseph Braun[4],
Sarah S. Comstock[5], Cristiane S. Duarte[6], Alison Hipwell[7], Hong Ji [8],
Janine M. LaSalle [9], Rachel L. Miller[10], Rashelle Musci[11], Jonathan Posner[6],
Rebecca Schmidt [12], Shakira F. Suglia [13], Irene Tung[7],
Daniel Weisenberger[14], Yeyi Zhu[15] & Rebecca Fry[16]

Environmental exposures, psychosocial stressors and nutrition are all potentially important influences that may impact health outcomes directly or via interactions with the genome or epigenome over generations. While there have been clear successes in large-scale human genetic studies in recent decades, there is still a substantial amount of missing heritability to be elucidated for complex childhood disorders. Mounting evidence, primarily in animals, suggests environmental exposures may generate or perpetuate altered health outcomes across one or more generations. One putative mechanism for these environmental health effects is via altered epigenetic regulation. This review highlights the current epidemiologic literature and supporting animal studies that describe intergenerational and transgenerational health effects of environmental exposures. Both maternal and paternal exposures and transmission patterns are considered, with attention paid to the attendant ethical, legal and social implications.

Heritability of health and disease risk in humans and other species involves influences beyond the genetic code. While many successes in large-scale human genetic studies exist, there remains a substantial amount of missing heritability in complex childhood disorders. That is, genetic variants have often explained a small fraction of the overall heritability of these diseases and their effect sizes are small. Environmental exposures, including chemical toxicants, psychosocial stressors, behaviors, and nutrition, are all potentially important

[1] Department of Preventive Medicine, Keck School of Medicine, University of Southern California, Los Angeles, CA, USA. [2] Department of Pediatrics, NYU Grossman School of Medicine, New York, NY, USA. [3] Department of Psychiatry, Boston Children's Hospital and Harvard Medical School, Boston, MA, USA. [4] Department of Epidemiology, Brown University School of Public Health, Providence, RI, USA. [5] Department of Food Science and Human Nutrition, Michigan State University, East Lansing, MI, USA. [6] Department of Psychiatry, Vagelos College of Physicians and Surgeons, Columbia University Irving Medical Center and New York State Psychiatric Institute, New York, NY, USA. [7] Department of Psychiatry, University of Pittsburgh, Pittsburgh, PA, USA. [8] Department of Anatomy, Physiology and Cell Biology, School of Veterinary Medicine, California National Primate Research Center, University of California, Davis, Davis, CA, USA. [9] Department of Medical Microbiology and Immunology, MIND Institute, Genome Center, University of California, Davis, Davis, CA, USA. [10] Icahn School of Medicine at Mount Sinai, New York, NY, USA. [11] Department of Mental Health, Johns Hopkins Bloomberg School of Public Health, Baltimore, MD, USA. [12] Department of Public Health Sciences, UC Davis School of Medicine, Davis, CA, USA. [13] Department of Epidemiology, Emory University, Atlanta, GA, USA. [14] Department of Biochemistry and Molecular Medicine, University of Southern California, Los Angeles, CA, USA. [15] Division of Research, Kaiser Permanente Northern California and Department of Epidemiology and Biostatistics, University of California, San Francisco, Oakland, CA, USA. [16] Department of Environmental Sciences and Engineering, Gillings School of Global Public Health, UNC Chapel Hill, Chapel Hill, NC, USA. ✉email: breton@usc.edu

influences that may impact health outcomes directly or via interactions with the genome. Such processes may explain some of the missing heritability for conditions that repeat across generations. Nonetheless, the hypothesis that factors other than the DNA sequence can be molecularly transmitted across one or more generations is contentious[1], eliciting well-worn Lamarck-versus-Darwin or nature-versus-nurture debates as well as raising concerns about epigenetic determinism, the risk of stigmatization or discrimination, and protection of privacy.

Some effects of environmental exposures can be transmitted across generations via changes to the sequence of germline DNA. It is also hypothesized that environmental exposure effects can be transmitted across generations via epigenetic modifications of germline DNA. For the purposes of this discussion, we will use the term intergenerational transmission to refer to transmission theoretically occurring when an exposure simultaneously affects the biology of an individual and its progeny, either by causing genetic or epigenetic changes to the individual and its germ cells or, in the case of pregnant females, causing genetic or epigenetic changes to the individual, its developing offspring, and the offspring's fetal germ cells. We will use the term transgenerational transmission to refer to transmission occurring when genetic or epigenetic changes are passed on to a subsequent generation that did not experience direct exposure either preconceptionally or prenatally (Box 1). When nonpregnant individuals are exposed, the earliest transgenerational effects that could be assessed are in their grandchildren; when pregnant females are exposed, the earliest transgenerational effects that could be assessed are in their great-grandchildren[2].

While evidence in support of transgenerational transmission exists in animal models[3–9], evidence in humans is virtually nonexistent, in part because of logistic, financial, and ethical hurdles that limit epidemiologic studies spanning multiple generations. Many common environmental exposures are not restricted to a single-generation in humans, making it difficult to distinguish between inherited and current effects[10]. Major challenges to demonstrating that epigenetic mechanisms are specifically responsible for transgenerational effects include the need to evaluate epigenetic marks across multiple tissues, the difficulties in ruling out confounding effects of genetic, ecological, and sociocultural factors, and the inability to control completely the effects of other environmental influences[11]. Additionally, for feasibility and ethical reasons, human studies often require minimally invasive biomarker collection, such as through saliva or blood, which is insufficient to capture the complex dynamics of epigenetic changes that may occur in specific tissues throughout development[12]. Last, there are ethical, legal, and social implications to this work that should be considered particularly in the context of structural inequities[13–15].

This review highlights the current epidemiologic literature and supporting animal studies that describe intergenerational and/or transgenerational health effects of environmental exposures. Both maternal and paternal exposures and transmission patterns are considered, with an emphasis on studies showing epigenetic modification as a potential mediator. Evidence for these processes in humans is drawn from the literature addressing a variety of exposures and health outcomes, including respiratory health, birth weight, obesity, cardiovascular health, and neurodevelopmental disorders. Last, we discuss the unique opportunities and challenges for investigating generational influences on child health in the Environmental influences on Child Health Outcomes (ECHO) study, which will integrate the study of genetics and epigenetics with exposures and health outcomes across diverse human populations in the United States.

**Genetic and epigenetic mechanisms.** Mechanisms of genetic inheritance include the following: germline chromosomal aneuploidies, germline DNA sequence variations such as single nucleotide polymorphisms, small segments of nucleotide insertions and deletions, and larger structural variants, as well as tandem repeats and retrotransposons (reviewed in Jackson et al. 2018[16]). Epigenetic inheritance involves mechanisms that do not change the DNA sequence but can still alter phenotypes and be passed on through mitosis or meiosis, depending on whether one is evaluating somatic or germline cells. Epigenetic modifications may include the attachment or removal of molecules to DNA itself or to the proteins (histones) around which DNA is wrapped when chromatin is bundled, the positioning of those bundles (nucleosomes), or the interplay of transcription factors and other enzymes within gene-regulatory networks. DNA methylation occurs when a methyl group is added to the 5-carbon of a cytosine ring, resulting in 5-methylcytosine (5-mC). This addition occurs most often in the context of a CpG site, which is a cytosine nucleotide located proximal to a guanidine nucleotide. Throughout the genome, the majority of CpG sites are methylated, with the exception of CpG islands in the germline and near promoter regions, which are largely unmethylated and associated with gene expression. MicroRNA (miRNA), small noncoding RNA (ncRNA), and transfer RNA (tRNA) also can exert post-transcriptional control over gene expression. Modifications of N-terminal tail amino acids in histones H3 and H4, such as methylation, acetylation, phosphorylation, and ubiquitination, are associated with open- and closed- chromatin states. Depending

---

**Box 1 ▌ Definitions of terms used**

Epigenetics
The study of heritable changes in gene function not due to changes in DNA sequence

Epigenetic inheritance vs genetic inheritance
The transmission of epigenetic marks across generations compared to the transmission of DNA sequences across generations

Intergenerational transmission
Transmission theoretically occurring when an exposure simultaneously affects the biology of an individual and its progeny, either by causing genetic or epigenetic changes to the individual and its germ cells or, in the case of pregnant females, causing genetic or epigenetic changes to the individual, its developing offspring, and the offspring's fetal germ cells.

Transgenerational transmission
Transmission occurs when genetic or epigenetic changes are passed on to a subsequent generation that did not experience direct exposure either preconceptionally or prenatally.[2]

DNA methylation
The addition of a methyl group ($CH_3$) to a cytosine base in DNA.

on location, histone methylation may enhance or suppress gene expression[17].

Higher levels of concordance in genome-wide methylation profiles between monozygotic vs. dizygotic twins suggest an interplay between genetics and epigenetics in the early years of life[18]. Using methylation arrays, McRae and colleagues[19] evaluated the role of genetic heritability in DNA methylation patterns across generations among families that included twin pairs, as well as their siblings and parents. Inherited similarities in DNA methylation profiles were largely due to genetic effects, and non-CpG sequence variation only accounted for 20% of interindividual differences. However, whole-genome bisulfite sequencing approaches for assessing DNA methylation in multiple tissues have demonstrated that population-variable CpGs are depleted in promoter regions, and interindividual DNA methylation differences are primarily nongenetic in origin, with a nonshared environment accounting for most of the variance[20]. More recently, next-generation sequencing-based approaches have identified regions of systemic interindividual variation that are inter-correlated over long genomic distances, conserved across diverse human ethnic groups, sensitive to the periconceptional environment, and associated with genes implicated in a broad range of human disorders and phenotypes[21]. Together, these investigations suggest that the combined role of environmental exposures and genetic variation on inherited DNA methylation patterns may depend on the genomic context.

**Germline changes associated with environmental exposures.** Transgenerational effects, which are hypothesized to occur across multiple generations and transcend direct exposure, require modification of the genome and/or epigenome of germline cells. For practical reasons, epidemiologic studies have primarily assessed outcomes of maternal prenatal exposure in the gametes of male offspring, using semen quality as an indicator of potential heritability. In one analysis, sons of overweight mothers were found to have higher sperm DNA fragmentation compared with sons of normal weight mothers[22]. Some investigations have found associations of birthweight, a proxy for the fetal environment, with sperm concentration, motility, and morphology[23–26]. Human studies have reported prenatal exposure to both persistent[27] and nonpersistent[28] environmental chemicals to be associated with altered semen parameters, findings that are supported by links between endocrine-disrupting chemicals (EDCs) and semen quality in the animal literature[29,30]. Taken together, these results suggest that in utero exposures may program male fetal germ cells and/or Sertoli and Leydig cells, which are laid down in the fetal testes and are responsible for future sperm production[31–33]. The consequences of such effects on future progeny are largely unknown.

While germline cells may be impacted by environmental exposures in utero, during childhood, and throughout reproductive life, the embryonic period of gonadal sex determination is the most sensitive to environmental insults influencing the epigenome, as this is when human primordial germ cells undergo extensive epigenetic modification[34]. Epigenetic changes to the male and female germlines may occur during gametogenesis in the adult gonads, as well[35]. In females, two major functions of the ovary are the production of germ cells and sex steroid hormones, chiefly 17β-estradiol and progesterone. At birth, the mammalian ovary contains a finite number of primordial follicles[36]. These follicles constitute the ovarian reserve from which preovulatory follicles develop. Postnatal exposure to environmental, industrial, chemotherapeutic, and xenoestrogenic chemicals can diminish ovarian follicles and alter steroidogenesis, resulting in impaired ovarian function and infertility in animal studies[37]. Animal and human studies have shown environmental conditions to affect oocyte maturation, potentially resulting in long-term developmental consequences. For example, psychological stress may reduce oocyte competence, and bisphenol A (BPA) exposure has been associated with adverse effects on oocyte maturation and embryo quality and implantation[38,39]. Even low concentrations of BPA alter the epigenome of mammalian female germ cells, impacting gene expression, chromosome dynamics, and oocyte development[40]. Within in vivo animal models, components of tobacco smoke have been shown to alter gonadotropin-releasing hormone 3 (gnrh3) DNA methylation, global 5mC levels, and global histone H3 lysine 4 dimethylation (H3K4me2) in zebrafish brains and oocytes, reducing their development and maturation capability[41,42].

In males, because sperm are produced continuously in the testes from puberty onward, current exposures are often considered as potential triggers of both genetic and epigenetic changes. A number of health conditions, behaviors, and environmental exposures in men have been associated with sperm DNA fragmentation, including obesity[43], smoking[44–46], varicocele[47,48], sexually transmitted infection[49–51], chemotherapy or radiotherapy[52], paracetamol[53], air pollution[54–57], and a variety of EDCs[58–61]. In turn, sperm DNA fragmentation has been linked to male factor infertility[62], miscarriage[63,64], and reduced live birth rate after in vitro fertilization[65]. Male offspring of subfertile fathers who used intracytoplasmic sperm injection to conceive were at increased risk of poor semen quality[66] and likely to have genetic abnormalities related to azoospermia[67], indicating the potential heritability of subfertility that may result from environmentally-induced alterations to germline DNA. Epigenetic mechanisms have also been proposed to explain the transmission of obesity from father to child[43,68]. Rodent studies indicate that, in addition to diet and exercise[69,70], paternal stress[71] and exposure to EDCs[72] may affect offspring health and behavior via epigenetic pathways. One of the most notable causes of sperm DNA fragmentation is oxidative stress (OS)[73,74]. Sperm are particularly vulnerable to genetic damage due to OS because sperm heads, which are filled with tightly packed chromatin, lack cytoplasm that contains the enzymes necessary for DNA repair[75]. OS may also affect the sperm epigenome, as some studies have shown hypoxia to be associated both with impairments in spermatogenesis and alterations in DNA methylation[76–81]. In general, it is theorized that OS-mediated sperm DNA fragmentation and/or epigenetic changes may underlie observed associations of a wide variety of paternal environmental and lifestyle factors with birth defects and childhood diseases[82,83].

## Rationale for epigenetic effects

*Single-generation epigenetic effects on children.* A growing body of literature supports associations between prenatal exposures and epigenetic modifications in children[84,85]. Epigenetic processes, such as CpG methylation and chromatin remodeling, are highly regulated during embryogenesis to ensure proper development[86,87]. Disruption of these mechanisms is of interest to child health researchers because they facilitate key developmental events, including placental and fetal growth, genomic imprinting, and cell differentiation[88]. While epigenetic modifications demonstrate plasticity during development, they also can remain stable, biologically embedding the effects of periconceptional exposures within offspring[89]. Environmental perturbation of epigenetic mechanisms can therefore change gene expression and result in altered outcomes, whether beneficial or adverse, at birth or later in life[89,90]. Nonetheless, studies that demonstrate epigenetic modifications that mediate the relationship between periconceptional exposures and child health using formal or causal mediation frameworks are generally lacking.

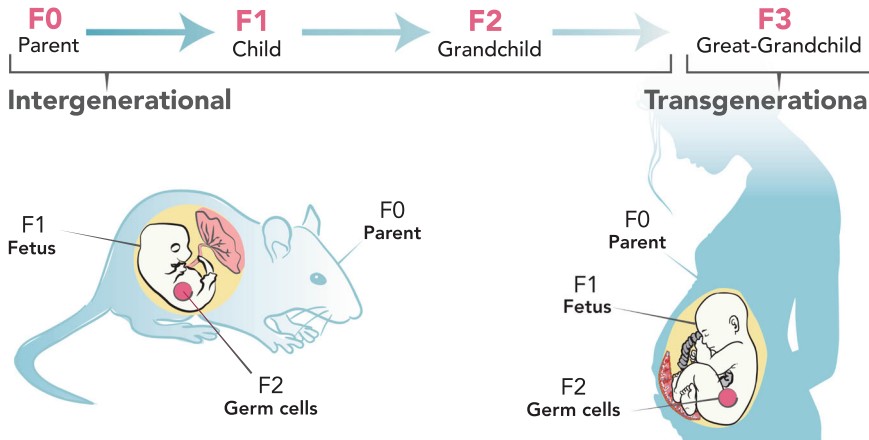

**Fig. 1 Intergenerational and transgenerational inheritance.** Depiction of inheritance patterns from the parent (F0) generation to the child (F1), grandchild (F2), and great-grandchild (F3) in humans and animals. An exposure in F0 can directly affect the developing fetus (F1) and the germ cells in F2; therefore, both routes of transmission are considered intergenerational. Transgenerational effects may be observed beginning with the F3 generation.

Epigenetic modifications may be induced by prenatal exposures and can be inherited intergenerationally, escaping the major waves of epigenetic reprogramming that occur during fertilization and gametogenesis[86,87,91–106]. One hypothesized molecular mechanism for bypassing the DNA methylation reprogramming wave is through small regulatory RNAs, sequentially generated in parental somatic tissues, packaged in extracellular vesicles (ECVs), and delivered to early embryos, where they ultimately drive a global reprogramming of genome expression[107]. Other means of escaping the early embryonic reprogramming are evidenced by CpG loci adjacent to intracisternal-A-particle elements or telomeric regions[106]. Single-generation epigenetic effects may also occur when exposures directly affect the developing somatic tissue postfertilization. Single-generation effects have been investigated primarily in relation to maternal factors, including diet, environmental chemical exposure, cardiometabolic disorders, gut microbiota, and mental health[90,108–113] Paternal preconception diet and exposure to drugs, toxicants, and EDCs have also been associated with epigenetic modifications among children[114,115].

Epidemiologic studies have often been limited in their ability to conclude causal effects of prenatal exposures on epigenetic modifications in children due to cross-sectional designs, underpowered analyses, cell-type heterogeneity, and other limitations common to epigenomic studies[104,116,117]. While increasing evidence supports that single-generation effects are likely to occur in humans, the extent to which these effects persist across multiple generations remains unclear.

**Animal models of epigenetic inheritance.** Numerous animal studies have described non-genetic inheritance of phenotypes, including the ones discussed below and others including eye color, cancer, and prostate and kidney diseases[3–9,118–123]. Often, these phenotypes result from environmental exposures during the parent generation, such as to particulate matter, diet, stress, and EDCs. This section discusses current evidence supporting both intergenerational and transgenerational effects on various health outcomes relevant to humans. In animal studies, generations are labeled as F0 (the exposed original generation), followed by F1, F2, F3… for subsequent generations. Because maternal F0 exposures that occur during pregnancy have the potential to affect the F1 fetus as well as the developing F1 gametes that give rise to the F2 grand offspring, only patterns of maternal

transmission that extend to the F3 generation meet the stricter criteria for transgenerational inheritance (Fig. 1).

*Respiratory health.* A number of rodent studies have focused on maternal prenatal environmental exposure and offspring respiratory outcomes. Some have investigated prenatal exposure to bacteria components, tobacco, nicotine, fungi, air pollutants, diet, maternal stress, and allergens in models of allergic asthma[124–133] (detailed models reviewed in Kraus-Etschmann et al. 2015[120]). Others have linked prenatal exposures such as environmental tobacco smoke, maternal immunization, and traffic-derived particulate matter to impaired lung function and/or structure[134–137], altered expression of asthma-related genes[138,139], or reduced allergen-specific IgE production[140]. A few studies included exposure during the preconception period, and two studies focused exclusively on this period[141,142].

Evidence for transmission of exposure effects on respiratory health-relevant outcomes beyond the F1 generation is emerging. Preconceptional and late-pregnancy sensitization to fungi was associated with lower IgE, altered airway eosinophilia, and differences in DNA methylation of IL-4 and IFN-g promoters in the F2 generation[128]. In utero exposure to environmental tobacco smoke altered lung function, increased inflammation, and was associated with changes in microRNA in both F1 and F2 generations[133]. In a study of maternal prenatal and perinatal exposure to phthalates in mice, increased allergic airway inflammation was observed in both F1 and F2 generations. A rat model of prenatal nicotine exposure demonstrated transmission of asthma phenotypes to the F1, F2, and F3 generation in males[143,127] and an association with epigenetic changes in reproductive and lung tissues in the F1 generation[127]. Exposure to environmental particles during mouse embryonic development increased asthma-like phenotypes in the F1, F2, and F3 generation in offspring[144]. Despite all of these findings, the underlying mechanisms for the inheritance of asthma risk via epigenetic processes remain poorly understood[120,145,146]. Causal links between transgenerational epigenetic changes and disease development have yet to be established.

*Obesity.* In animal models of rats and mice, maternal prenatal nutrition has been shown to influence offspring development and long-term health, including body size, obesity, and related metabolic disorders[147–149], although the mechanisms remain unclear[10,150,151]. Consumption of a high-fat diet during

pregnancy has been shown to increase rates of obesity and metabolic syndrome in rats by predisposing offspring to an obesogenic phenotype[152], providing some evidence of developmental programming that has been shown to persist across generations in studies of other rodents and mammals[153–155]. Few studies have examined F3 generation phenotypes, although increased body size and related phenotypes have been shown to persist across two generations of mice after prenatal exposure to a maternal high-fat diet[156], with sex-specific and parental lineage effects in the F3 offspring[157].

As previously reviewed in Vickers 2014[119], there is additional evidence that ancestral exposure to obesogenic chemicals results in the transmission of obesity-related phenotypes through at least three generations[158]. Exposure of dams to EDCs such as the pesticide dichlorodiphenyltrichloroethane (DDT), a mixed hydrocarbon jet fuel (JP-8)[159], the herbicide glyphosate[160], and common plastic ingredients such as BPA, diethylhexyl phthalate, and dibutyl phthalate[72], has led to increased obesity in F3 rats[161]. Prenatal exposure to environmental obesogens, including tributyltin[162,163], heavy metals[164], and the pesticide methyoxychlor[165], has been shown to produce lasting effects to at least the F4 generation.

There is also some evidence that inherited sperm epigenetic mutations (epimutations) underlie transgenerational obesity as well as other outcomes in the F3 generation following F0 exposure during pregnancy[72,166,167]. Observed alterations to IGF-2 and H19 expression in F1 sperm and F1 and F2 offspring suggest their involvement, along with other sperm epimutations[168], in the transmission of the obesity phenotype[169]. Transgenerational effects of obesogenic exposures could also occur through alterations in chromatin structure and accessibility[163,170] or other mechanisms[171].

*Neurological/behavioral outcomes.* As previously reviewed[118,121,122], many animal studies provide evidence for epigenetic and long-term neurodevelopmental and behavioral consequences of maternal prenatal stress[172], nutrition, obesity[173], and other exposures[174] in the developing offspring. Other studies suggest that paternal exposures can alter offspring neurodevelopment, potentially through epigenetic changes in sperm[172,175–179]. Fewer studies have investigated whether the effects of parental exposure are passed on to F2 generations and beyond. In zebrafish, prenatal exposure to contaminants, including low-dose inorganic arsenic[180] and lead[181], produced neurobehavioral, epigenetic, and brain transcriptomic alterations in the F2 generation. In mice, early postnatal stress transmitted depressive-like behaviors up to the third-generation (F1, F2, and F3). Behavioral traits co-segregated with altered DNA methylation in the male germ line[182,183], and there was evidence for microRNA involvement[176,177], suggesting that transgenerational inheritance could occur through the transmission of acquired epigenetic marks. Evidence in *C. elegans* suggests that small RNAs in neurons control chemotaxis behavior of the progeny for at least three generations[184]. Prenatal infections and immune challenges that produce intergenerational neurodevelopmental and neurobehavioral disorders have also been shown to transmit transgenerational behavioral abnormalities to F3 offspring, primarily through the paternal lineage and potentially through overlapping epigenetic mechanisms[185]. Rodent studies of impaired maternal care and social context produced altered behaviors, including neophobia in F1[186] that persisted to the F2 and F3 generations through epigenetic mechanisms[187]. Maternal high-fat diet led to cognitive disabilities and an altered response to a noncompetitive receptor antagonist of the N-methyl-D-aspartate receptor in adult mouse F2 and F3 offspring in a sex-specific manner[188]. Maternal BPA exposure disrupted social interactions in F3 mice and dysregulated normal expression of postsynaptic density genes during neural development[189]. Similarly, chronic paternal adolescent stress produced transgenerational behavior and amygdala transcriptomic changes in F1 and F2 mice[190].

**Evidence in humans**. Human studies that document both intergenerational and transgenerational effects of environmental exposures on health outcomes are relatively rare. This section presents available evidence, covering a range of environmental exposures and health outcomes. The underlying mechanisms that might explain these effects have been neither uniformly nor comprehensively tested and are, in fact, the subject of some debate[1,2,11]. While the focus here is on hypothesized epigenetic mechanisms of intergenerational and transgenerational transmission of environmental exposure effects, we acknowledge and discuss the evidence and the challenges in disentangling potential epigenetic mechanisms from concurrent genetic, ecological, and sociocultural factors that may track across generations.

*Inter- and transgenerational effects on respiratory health.* Moderate evidence exists for transmission of asthma, allergy, and other respiratory diseases through the third-generation. In the nationally representative Child Development Supplement to the Panel Study of Income Dynamics cohort of more than 2500 children, children with a grandparent with asthma had 1.5 times the odds of reported asthma. Children with a parent and grandparent with asthma had over four times the odds of reported asthma compared with those without a parental or grandparental history of asthma[191]. Studies such as these have implicated genetic risk factors or gene-environment interactions for inherited asthma risk across a generation;[192–195] however, nonbiological explanations and interactions between genetic and non-genetic risk factors are also feasible. For example, multiple generations of families living in low-socioeconomic neighborhoods with higher exposure burden and greater poverty may also explain or exacerbate asthma risk[196,197].

Perhaps, the most well-studied environmental risk factor for asthma is tobacco smoke. A number of cohort studies have begun to evaluate inter- and transgenerational asthma risk to cigarette-smoking exposure. In a case-control study nested within the Children's Health Study in Southern California, grandmaternal smoking during pregnancy was associated with a 2-fold increased odds of asthma in the grandchild[198]. If both the child's mother and grandmother smoked during pregnancy, the child had even higher odds of developing asthma compared with no exposure[198]. In the Norwegian Mother and Child Cohort Study, grandmaternal smoking during pregnancy was associated with 15% and 21% greater relative risk of child asthma at ages 3 and 7 years, respectively, and 15% greater relative risk for dispensed asthma medications at age 7 years[199]. Data from the Ageing Lungs in the European Cohorts Study found that grandmaternal smoking during pregnancy was associated with child asthma with nasal allergies[200]. In comparison to these studies based on retrospective reporting, prospectively collected data from the Swedish National Board of Health and Welfare and Statistics Sweden registries demonstrated that children of age 1 to 6 years had increased odds of asthma or of wheeze or asthma if their grandmothers smoked during weeks 10–12 of pregnancy, regardless of maternal smoking history. The odds of asthma were higher with greater cigarette exposure (10 + cigarettes/day vs. none: OR 1.23, 1.17–1.30). Grandmaternal smoking was associated most robustly with child early persistent asthma compared with early transient and late-onset asthma[201]. In contrast, the English population-based Avon Longitudinal Study of Parents and Children did not find evidence of transmission of effects of grandmaternal smoking during pregnancy on child lung function, bronchial responsiveness, or doctor-diagnosed asthma[202]. Overall, these studies provide relatively strong epidemiological support for an association between grandmaternal smoking and child asthma outcomes. Notably, because most of this research relied on retrospective reporting of exposures, unmeasured residual

confounding, and potential recall bias may have influenced results. For example, differences in maternal education, geographical and temporal trends in smoking patterns, income level, and race and ethnicity not sufficiently included in study samples or completely accounted for in covariate-adjusted models may contribute to some of the discrepancies in the results.

A few studies also have compared differences in child asthma risk depending on maternal versus paternal line of transmission to examine whether fathers' smoking may influence risk through, presumably, heritable epigenetic alterations in sperm precursor cells prior to conception; however, the results are mixed[202,203]. While some emerging evidence implicates epigenetic regulation as the underlying observed transgenerational health effects, the findings have been inconsistent[204,205]. Cross-generational effects in the absence of direct exposures may induce apparent changes in disease risk among generations through several mechanisms beyond epigenetic ones, including shared familial environment or other cultural effects[146,204].

*Inter- and transgenerational effects on birthweight.* A few studies have examined birthweight across three generations. In the Uppsala Birth Cohort Multigenerational Study in Sweden, correlations between grandparent and grandchild birthweight were stronger along the maternal line; however, this finding did not account for maternal birthweight[206]. In the Aberdeen cohort, grandmother's birthweight was associated with child birth weight independent of maternal birth weight as well as prenatal and sociodemographic factors[207]. Overall, the data from these and other cohorts suggest that social, environmental, and metabolic factors experienced by grandparents influence their grandchildren's birthweight. Whether these effects are mediated by epigenetic mechanisms is unknown. In some cases, epigenetic modifications, such as DNA methylation, have been associated with birthweight in the F1 generation, but their persistence to the F2 generation or beyond has not been assessed[100,208–210]. Although the methylation patterns that are associated with birthweight persist into childhood, there is evidence that they may not persist into adulthood[100,211]. Therefore, transfer to the F2 generation is less likely, although possible, since these human studies do not assess the methylation of gamete cell DNA. Methylation patterns at the PBX1 locus have been associated with birthweight in more than one cohort, but are not associated with others[210–212]. This discrepancy in the results is unfortunately common in intergenerational epigenetic studies and may be due to differences in the environmental exposures that establish the epigenetic patterns.

Current data suggest that some of the intergenerational social influences on birth weight include education, geographical location, and other sociodemographic characteristics. For example, grandparents' educational attainment and residential environment have been associated with grandchildren's birthweight[213,214]. In the Fragile Families and Child Wellbeing Study, having a grandfather with less than a high school education was associated with a 93-gram reduction in birth weight and a 59% increase in odds of low birthweight[214]. An examination of multiple social and biological factors within the Aberdeen Children of the 1950s study revealed that both distal and proximal grandparental and parental life-course biological and social factors predicted child size at birth. Inequalities in size at birth appeared to be generated largely via continuity of the social environment across generations, with inequalities in maternal early-life growth predicted by the grandparental social environment during the mother's early life[215]. Studies of the social influences of intergenerational effects on birthweight rarely address potential epigenetic mechanisms by which these exposures affect outcomes.

BMI and smoking history across generations have also been examined for their relations to grandchildren's birth weight, but

studies have largely failed to find direct associations. Some data suggest that associations between grandparents' BMI and grandchildren's birthweight are mediated by maternal BMI[216,217]. There is evidence that smoking may have intergenerational effects on birthweight. One study, which accounted for family clustering, found that birth weight was higher in children whose grandmother and mother both smoked during their pregnancies relative to children whose grandmother and mother both did not smoke during pregnancy[218]. This association was dependent on the grandmother's birth cohort: infants whose grandmothers were born between 1929 and 1945 experienced this effect, but infants whose grandmothers were born between 1904 and 1928 did not. Other studies have also shown that the relation between grandmothers' smoking and children's birthweight is mediated by maternal smoking[217,219,220]. However, the pattern of the results of these studies is not consistent. Notably, these analyses have been conducted across a number of different time periods, when the content of cigarettes and the intensity of their use were changing frequently. Therefore, the transgenerational influence of this environmental exposure on birthweight, and any role for epigenetics, remains unclear.

*Inter- and transgenerational effects on cardiovascular health.* Changes in parents' and grandparents' food access have been associated with children's cardiovascular health, sometimes with sex-specific patterns. Analysis of data from three cohorts born in 1890, 1905, and 1920 in Overkalix Municipality in northern Sweden revealed that if the food was not readily available during the father's slow growth period (ages 9 to 12 years), then the child's cardiovascular disease mortality was low; diabetes mortality increased if the paternal grandfather was exposed to a surfeit of food during his slow-growth period (OR 4.1, 95% CI 1.33–12.93)[221]. Cardiovascular mortality in the grandchild was not associated with sharp changes in food access experienced by maternal grandparents or paternal grandfather. However, if the paternal grandmother lived through a sharp change in food supply from year to year prior to puberty, her sons' daughters had an excess risk for cardiovascular mortality (OR 2.69, 95% CI 1.05–6.92)[222]. Other studies have noted associations between grandmother's exposure to famine during pregnancy and grandchild's development of poor cardiometabolic outcomes by adulthood, including hyperglycemia and type 2 diabetes[223]. In the case of the Dutch Hunger Winter, individuals prenatally exposed to famine also had less DNA methylation of the imprinted IGF2 gene compared with their unexposed, same-sex siblings six decades later. These data suggested that very early mammalian development is a crucial period for establishing and maintaining epigenetic marks[224].

*Inter- and transgenerational effects on neurodevelopmental outcomes.* Numerous grandparental exposures have been linked to grandchild neurodevelopmental outcomes. This section summarizes studies that have examined third-generation effects of tobacco and alcohol use, and stress/trauma on child neurodevelopmental outcomes, including cognitive abilities, emotional and behavioral symptoms/diagnoses, neurodevelopmental disorders, and substance-use disorders.

As was the case for respiratory health, grandparental smoking has also been associated with neurodevelopmental effects in the grandchildren. Existing studies have focused largely on associations with child-externalizing problems (e.g., aggression, rule-breaking). In at least two studies, the association between grandparental smoking history or substance use and child-externalizing behaviors was mediated by parental psychological symptoms and tobacco use[225–227]. Other studies have examined transgenerational effects of smoking on attention-deficit/

hyperactivity disorder and an autism-spectrum disorder. An analysis of the Norwegian Mother and Child Cohort found that grandparental smoking during pregnancy had a similar magnitude of association with child attention-deficit/hyperactivity disorder diagnosis as maternal smoking during pregnancy[228]. These findings suggest that the association between maternal smoking during pregnancy and child attention-deficit/hyperactivity disorder may not be due to causal intrauterine effects, but rather reflect unmeasured confounding factors. In the Avon Longitudinal Study of Parents and Children, maternal grand-maternal smoking during pregnancy was associated with elevated granddaughter scores on autism spectrum traits and increased risk for autism-spectrum disorder diagnosis[229]. Notably, the effects were magnified for children whose mothers did not smoke during pregnancy. In contrast, paternal grandmaternal smoking was not related to these outcomes in grandchildren[229].

Fewer studies exist on the effects of alcohol across generations. Available evidence suggests that grandparental and parental alcohol use have independent, cumulative effects on child risk for alcohol-use problems[230,231]. A large survey of college students revealed that problem drinking was greater among students reporting a parent or grandparent diagnosed with or treated for alcoholism, with the highest rates among those for whom both a parent and grandparent were affected[230]. Grandparental alcohol-use problems appear to have unique effects on child alcohol use problems, even when the parent is apparently unaffected[231]. In a nationally-representative sample of adults in Sweden born between 1980 and 1990, stability in alcohol-use disorders (AUD) declined by ~50% between the first and second generations. However, grandchildren with two or more grandparents affected by AUD had 40% greater AUD risk relative to grandchildren with only one affected grandparent[231]. Limited evidence suggests that the effects may be moderated by sex, with stronger associations of grandparent and parent AUD with child AUD when generations are of the same sex[231]. Across studies, the cross-generational patterns suggest a possible role for genetic transmission of risk for AUD; however, studies that consider the degree of exposure to alcohol-abusing family members indicate that risk is at least partially socially mediated[232]. Grandparent alcohol use also may influence grandchild behavioral precursors known to increase the risk for alcohol-use problems, such as aggression and inhibitory control. For example, one study found that grandparent alcohol use was associated with child aggression, with the effect mediated by parental AUD and by marital and parent–child aggression[233]. However, another study failed to find an association between grandparent alcohol use and child inhibitory control[234]. More research in this area is needed, including designs that can disentangle the independent and joint influences of genetic, epigenetic, social, and environmental factors on cross-generational associations. Notably, when interpreting findings in this area, consideration of the impact of changing norms about alcohol use across generations, including types of alcohol consumed and motivation for drinking, is important[235].

In addition to the effects of chemical exposures, stress and trauma have also been evaluated across generations for their effects on neurodevelopment. The majority of third-generation studies examining the putative effects of grandparental stress and trauma exposures have been conducted among Holocaust survivors. In a meta-analysis of 13 community samples representing 1012 children with or without a grandparent who was a Holocaust survivor, Sagi-Schwartz and colleagues concluded that there was no evidence for tertiary traumatization[236]. Specifically, there were no differences in the psychological well-being and adaptation between grandchildren of Holocaust survivors and comparison groups; this result held regardless of the manner of participant recruitment. The results highlight the notion that Holocaust survivors may represent a group of resilient individuals without a genetic bias to developing posttraumatic stress reactions; some suggest such a characteristic helped promote survival (consistent with work on Rwandan survivors described by de Quervian et al. 2007[237]) and, subsequently, facilitated the well-being of their children and grandchildren[238]. Consideration of other contributing factors among Holocaust survivors is important given evidence that they tend to have fewer children per family than comparison groups[239]. Finally, one study has examined tertiary trauma risk in a non-Holocaust survivor sample of 2,282 first-grade children from seven districts in Shanghai, China[240]. Analyses showed that children whose parents and/or grandparents experienced traumatic events had higher scores on parent-reported behavioral and emotional problems compared with children with no family trauma history. However, no differences were observed when teacher reports were examined. The likely influence of parental characteristics on report of child behavior was further indicated in analyses that accounted for parental depressive symptoms, physical health, and parenting factors; in these adjusted models, any effect of grandparent trauma on child emotional and behavioral outcomes became negligible.

A handful of studies also show alterations to DNA methylation from parental-exposed trauma. For example, post-traumatic stress disorder from exposure to the Tutsi genocide has been associated with NRC31 epigenetic modifications in both mothers and their offspring, suggesting alterations to the HPA axis[241]. Adult offspring of Holocaust survivors show reduced DNA methylation and increased gene expression in FKBP5, a gene associated with indices of glucocorticoid sensitivity[242].

To date, current evidence does not provide consistent support for third-generation effects of stress exposures on child neurodevelopmental outcomes. However, caution is warranted in making firm conclusions from the extant literature. Studies are relatively few in number and vary widely in sampling design, life stage of assessment, and type of outcomes measured. Retrospective and proxy reporting of exposures by parent or offspring may also be subject to bias.

**Current operational and analytical challenges in the field.** Inter- and transgenerational epidemiological studies face several obstacles to their success (Box 2). One of the greatest operational challenges to performing intergenerational and transgenerational investigations in human populations is the lengthy time (20–40 years) between generations. As a result, there is a paucity of data on the fourth generation in humans. This inherent limitation poses challenges to cohort retention and creates problems in terms of technological changes related to biospecimen storage and medical record keeping. Retrospective collection of information about experiences, exposures, and medical conditions is subject to both ascertainment and recall biases, particularly if a child or grandchild has a medical condition. Furthermore, parental age can be highly variable within and between families and generations of a human study, meaning that generations being compared within large human studies may span multiple decades. Methods for controlling for these inherent time variables are being developed, but this is an area where future research could be beneficial. Animal models that test specific hypotheses related to human intergenerational and transgenerational studies are clearly important to verify any conclusions because they can be conducted faster and can control for generation time, age, genetic background, and environment.

A second major operational challenge is that existing studies vary widely in the specific exposures and outcomes assessed

---

**Box 2 | Major challenges for epidemiologic studies or inter- and transgenerational effects**

- A long time period is needed to observe changes across generations in human studies, during which time study leadership, methods, staffing, and funding may change.
- Methods for phenotyping health outcomes or exposure assessment are often not standardized, or standards may evolve over time, making it hard to compare across cohorts and over time.
- In addition to shared genetics, families tend to have shared living environments and shared environments such as socioeconomic status, diet, tobacco and alcohol use, and other cultural behaviors. This makes disentangling shared genetics from a shared environment a challenge.
- There is a need to balance individual responsibility for health and structural or societal responsibility for health. Epigenetic programming raises concerns over the risk of stigmatization or discrimination broadly as well as specifically regarding parenting and reproduction, legal proceedings, political theory, and privacy concerns.
- The level of privacy protection needed for epigenetic research is currently debatable.
- Integrating data from multiple 'omics layers across multiple generations would help to obtain the most holistic investigation of disease risk inheritance within families, but statistical methods for doing so are complicated, computationally challenging, and continually evolving.v
- Epigenetic marks can vary by cell type or subtype. Because most human epigenetic studies collect biological samples from living individuals, tissue sources are predominantly blood, saliva, buccal, and semen collections. Care must be taken to interpret the relevance of changes in these surrogate tissues to human health.

---

across generations and geographical locations, and in the study designs used to collect and analyze the data. Clinical assessments and phenotyping of outcome measures need to be standardized with respect to the developmental stage within a study when possible. For many environmental chemical exposures, the most valid, uniform method may be to measure them in biospecimens; however, this requires that cohorts banked relevant biospecimens generations ago, that the biospecimens were stored in a manner consistent with current best practices for given assay requirements, and that the exposures of interest do not degrade over such a long period in storage. Moreover, there may be ethical concerns about using samples from subjects whose informed consent was provided years or decades ago and may not directly address these types of assays in biospecimens. Furthermore, different study designs and/or analysis strategies can lead to different conclusions from the data[243].

Epigenetic marks are potentially uniquely poised to serve as the interface between genetics and environment, but careful interpretation of epigenetic results from intergenerational and transgenerational studies is critical. Evidence that an epigenetic mark is inherited within a family does not necessarily mean that it is purely environmentally induced and epigenetically transmitted. For example, some single nucleotide polymorphisms may determine DNA methylation if they cause a gain or loss of a CpG dinucleotide. These are expected to be rare (< 1%), based on a study of neonatal genotype-methylation comparisons[244]. However, even when genetic analyses of common single nucleotide polymorphisms are included in cross-generational epigenetic study designs, genetic effects from rare variants or copy number variants may remain that can impact the interpretation of the results. Ideal intergenerational and transgenerational epigenetic study designs would include the integration of multiple 'omic layers of information to obtain the most holistic investigation of disease risk inheritance within families. Multiomic study designs offer many benefits, including increased confidence in the results when specific genes or gene pathways are identified in more than one layer of information. In addition to shared genetics, families tend to have shared living environments, socioeconomic status, diet, tobacco and alcohol use, and other cultural behaviors. Structural inequities may persist within families across generations, raising concerns that the epigenetic marks may be erroneously considered heritable if inequities are not adequately captured or measured. For example, socioeconomic status may be a predictor of certain environmental

exposures as well as many health outcomes and can persist across generations[245]. Controlling for such confounding requires appropriate study designs that can mitigate the problem (e.g., sibling designs) or adequate measurement of potential confounders. Second, the timing of environmental exposures in the grandparent's generation relative to their own and their children's gamete development must be considered.

A remaining challenge is the choice of tissues for collection and investigation. Unlike DNA, which is consistent across tissue sources, epigenetic marks can vary by cell type or subtype. Because most human epigenetic studies collect biological samples from living individuals, tissue sources are predominantly blood, saliva, buccal, and semen collections. Determining whether these surrogate tissues are relevant for investigations of less accessible tissues, such as brain or other internal organs, may require comparisons of existing databases of epigenetic roadmaps of multiple human tissues and cell types relevant to the disease of interest[246]. In addition, correction for cell-type composition is important in human epigenomic studies to determine if the epigenetic changes being investigated are a result of a cell composition change, as opposed to changes within all cell types or a specific cell lineage[247]. Because environmental exposures and gene-by-environment interactions that occur *in utero* have a particular impact on long-term health outcomes, samples obtained from women and offspring during pregnancy and at birth, including maternal and cell-free fetal DNA from maternal prenatal plasma, as well as the placenta, cord blood, and newborn blood spots, should be particularly useful for intergenerational epigenetic studies involving childhood diseases[248].

**Ethical, legal, and social implications of epigenetic research.** The complexity of epigenetic programming of health highlights the inherent tension in the balance between individual responsibility for health and structural or societal responsibility for health[15]. The current field of epigenetics lies at the crossroads of ethics, law, and society, having stimulated much discussion in recent years and yielding diverse opinions regarding the balance between risks and rewards of epigenetic studies[13–15]. Epigenetics blurs the line between the interconnectedness of nature and nurture in the inheritance of traits and disease susceptibilities within families. This leads to concerns about new forms of epigenetic determinism or environmental determinism based on the possibility of transgenerational epigenetic effects on child

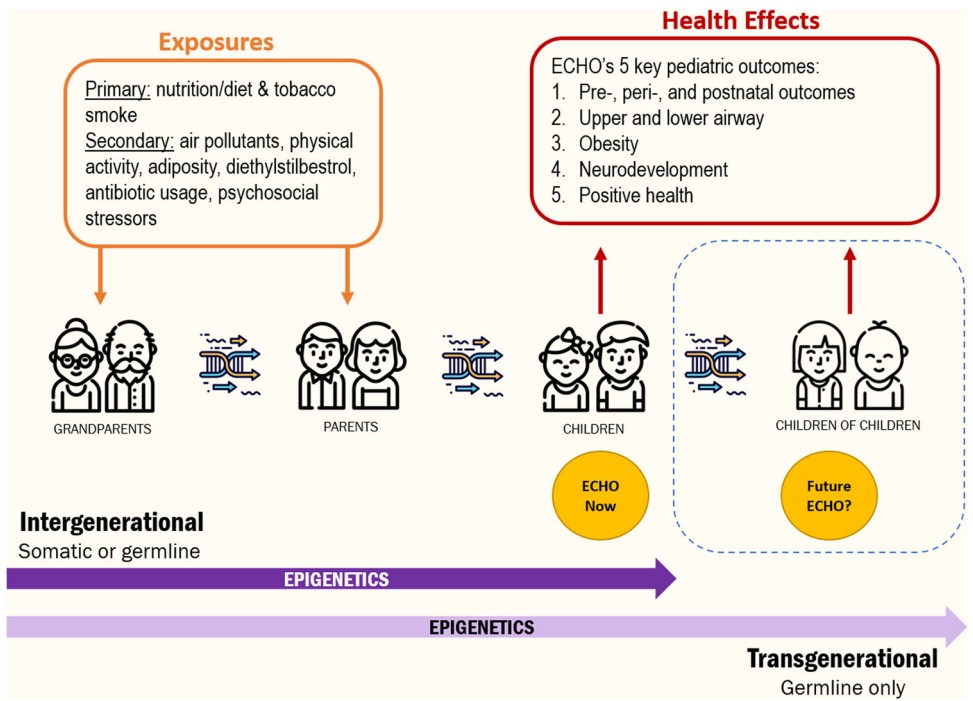

**Fig. 2 Opportunities in ECHO for understanding the influences of the environment on child health across generations.** The Environmental influences on Child Health Outcomes (ECHO) program, created in 2016, is an NIH-funded national program that supports existing observational and intervention studies to answer important questions about key developmental areas, including pre-, peri-, and early postnatal outcomes, airway conditions, obesity, and neurodevelopment. Exposures of interest include nutrition, tobacco smoke, pollutants, physical activity, drug usage, and stress. The ECHO program offers a unique platform and opportunity in which to explore the health effects of exposures, including more contemporaneous ones, that are present or have far-reaching consequences across generations.

development. Scientists have raised concerns about the "biologization of social space and time" that is, that environmental and socio-cultural circumstances conceptualized as external factors can be internalized into the body. Adverse effects or unintended consequences may arise "from translating social disparities into biological inequalities and reducing complex social problems to molecular codes"[13]. However, a potential advantage of understanding biological correlates of social inequities is that the impact of these inequities on health could be measured before and after interventions, opening up the possibility for epigenetic biomarkers to become an important part of medical care in the future.

Other areas of concern involve the risk of stigmatization or discrimination broadly as well as specifically regarding parenting and reproduction, legal proceedings, political theory, and privacy concerns. The focus on developmental origins of adult disease potentially places undue scrutiny on the behavior of pregnant women, which could lead to greater control over the female body[249]. For instance, could epigenetic data be used in legal proceedings to support claims of negligent parenting?[250] Challenges also arise when defining a reference epigenome which epigenome should be considered the healthy reference? Will a new and greater epigenetic understanding of social disparities help target interventions or will it give rise to increased racialization without structural determinants of inequities being addressed? An additional concern is that newly developed epigenetic norms may unduly burden the already-marginalized groups by placing more intense scrutiny and blame for sociocultural practices or behaviors that might affect the new epigenetic norm[13], rather than focusing on upstream, structural factors that produce health disparities.

Last, there is an ongoing debate over the level of privacy protection needed for epigenetic research. On the one hand, the genetic privacy framework may be adequate and translatable to epigenetics, while on the other hand epigenetic data may be more sensitive and require greater protection[14]. At the heart of the epigenetic debate lies the varied attempts by scientists and bioethicists to account for the social, political, economic, and cultural context of our biology[15].

**An opportunity: the ECHO study.** The Environmental influences on Child Health Outcomes (ECHO) program, created in 2016, is an NIH-funded national program that supports existing observational and intervention studies to answer important questions about key developmental areas, including pre-, peri-, and early postnatal outcomes, airway conditions, obesity, and neurodevelopment[251]. The program consists of 72 cohorts (https://www.nih.gov/echo/pediatric-cohorts), with an estimated sample size exceeding 50,000 children from diverse populations across the United States. The program leverages extant data along with newly collected data from primary and secondary sources along with biospecimens collected across key developmental periods[252]. The structure of ECHO is such that the principal investigators of each cohort form a steering committee designed to guide the research program. The ECHO program is also supported by a funded coordinating center, data analysis center, children's health exposure analysis resource, patient-reported outcome core, and genetics core[253]. ECHO forms a large multi-disciplinary network of researchers focused on solution-oriented research among a heterogeneous set of existing cohorts. This structure differs from other large-cohort studies (e.g., Avon Longitudinal Study of Parents and Children) in that the extant cohorts that are a part of the ECHO program vary significantly in their study design, year of initiation, research domain focus, and study population. This heterogeneity offers unique challenges related to the harmonization of extant data, but also provides

researchers with the diversity needed to answer important questions regarding the impact of biology, structural inequities, changing policy, and socio-cultural context on child health.

While inter- and transgenerational data collection remains challenging and the ethical, legal, and social implications of such research must be considered, the ECHO program offers a unique platform and opportunity in which to explore the health effects of exposures, including more contemporaneous ones, that are present or have far-reaching consequences across generations (Fig. 2). These exposures, which are extensively measured within the ECHO cohorts, are hypothesized to impact a number of health outcomes in children. As a part of ECHO, biological specimens, which can be used for genetic and epigenetic analyses, are being collected and analyzed. When considering maternal–paternal–child trios for intergenerational hypotheses, we estimate that approximately 10,000 will be available in the ECHO program. Further, the possibility of extending the ECHO program to include a second-generation expands the potential reach of ECHO in a way few studies have been able to accomplish. Indeed, several cohorts within ECHO are already recruiting and collecting data on this third-generation. With the availability of these large numbers of dyads and triads and the potential for the collection and analysis of epigenetic, genetic, and 'omics data in combination with socio-economic and cultural data, the ECHO program is poised to answer key questions regarding the mechanisms through which environmental exposures impact child health across multiple generations.

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

## Acknowledgements

The authors wish to thank our Environmental influences on Child Health Outcomes (ECHO) colleagues, the medical, nursing, and program staff, as well as the children and families participating in the ECHO cohorts. We also acknowledge the contribution of the following ECHO program collaborators: Coordinating Center: Duke Clinical Research Institute, Durham, North Carolina: Smith PB, Newby KL, and Benjamin DK. Research reported in this publication was supported by the Environmental influences on Child Health Outcomes (ECHO) program, Office of The Director, National Institutes of Health, under Award Numbers U2COD023375 (Coordinating Center), U24OD023382 (Data Analysis Center), UH3OD023287, UH3OD023305, UH3OD023337, UH3OD023313, UH3OD023285, UH3OD023328, UH3OD023244, UH3OD023342, UH3OD023365, UH3OD023282, UH3OD023289, and UH3OD023348 This work was also supported by the following NIH grants: 5K99ES030403, R01MH121070, R01AI141569-1A1, P30ES023513-supported EHSC scholar fund, R01ES029213, R01HD093643, R01HL125761, T32AA00745, and K01DK120807 The content is solely the responsibility of the authors and does not necessarily represent the official views of the National Institutes of Health.

## Author contributions

CVB conceived of the concept for the review paper, facilitated the organization of review sections and coordination of the writing team, and helped write the introduction, ethics, challenges, and conclusion sections. RL facilitated organization of review sections and coordination of the writing team and served as editor. LGK wrote the section on germline changes associated with environmental exposures and overall editing. MBE provided overall editing and streamlining. AKP wrote the introduction. TB provided overall editing. JB wrote the section on neurodevelopmental outcomes in humans. SSC wrote the section on inter- and transgenerational effects on birthweight. CSD contributed to overall editing and the neurodevelopment section. AH wrote the section on neurodevelopmental

outcomes in humans. HJ wrote the section on evidence in animals. JML wrote the introduction and the section on challenges to the field. RLM wrote the section on respiratory health in humans. RM wrote the section on opportunities in ECHO. JP provided overall editing. RS wrote the section on evidence in animals and on opportunities in ECHO. SFS wrote the section on birthweight in humans and cardiovascular effects. IT wrote the section on neurodevelopmental outcomes in humans. DW wrote the section on rationale for epigenetic effects. YZ provided overall editing. RF wrote the sections on germline changes and single-generation effects in humans, created Fig. 1, and provided overall editing.

## Competing interests
Joseph M. Braun's institution was financially compensated for his services as an expert witness for plaintiffs in litigation related to PFAS-contaminated drinking water; these funds were not paid to him directly. All other authors declare no competing interests.
