## [Peer Review File · Communications Biology]

Reviewers' Comments:

Reviewer #1:

Remarks to the Author:

In this manuscript, Breton et al. compiled vast amounts of epidemiology literature to provide a comprehensive summarization of the inter- and trans-generational effects of environmental exposures on health outcomes in both animals and humans. Overall, this is a well-organized and insightful review. The authors covered a very diverse set of environmental factors and health outcomes, and considered both maternal and paternal exposures, thus providing an excellent resource for this complicated topic. The manuscript focused heavily on epidemiology literature that provides clinical observations to associate environmental exposures to inheritable health outcomes, yet lacking in mechanistic understanding of the process and its relation to epigenetics. However, these are known limitations and challenges in the field and are well-acknowledged by the authors.

The following suggestions may strengthen this manuscript:

1. In "Higher levels of concordance in genome-wide methylation patterns between monozygotic vs. dizygotic twins suggest an interplay between genetics and epigenetics." (page 4), it would be more accurate to state that this high concordance of methylation patterns in monozygotic twins appears in the early years of life according to study by Fraga et. al. 2005. PNAS. PMID: 16009939.
2. In "OS-mediated sperm DNA fragmentation and/or epigenetic changes may underlie observed associations of a wide variety of paternal environmental and lifestyle factors with birth defects and childhood diseases" on page 5, what does "OS-mediated" refer to? Maybe ROS-mediated? Besides, one study reported that impairments of spermatogenesis (e.g. decrease in sperm count and sperm motility) that induced by hypoxia exposure are associated with methylation patterns (Wang et. al. 2016. Nat. Commun. PMID: 27373813), which should be considered.
3. In the section "Inter- and transgenerational effects of birthweight", the authors have reviewed the transgenerational influences of social, environmental, and metabolic factors on the birth weight. It might be necessary to discuss the potential epigenetic mechanisms that transmit these effects to birth weight of generations as reported in these studies, e.g. Küpers et. al. 2019. Nat. Commun. PMID: 31015461; Lin et. al. 2017. BMC med. PMID: 28264723; Agha et. al. 2016. Clin Epigenetics. PMID: 27891191.
4. Reference 17 does not seem to contain any information regarding epigenetics, might consider removing
5. Reference 98 does not seem to contain any information regarding to epigenetics, might consider removing

Reviewer #2:

Remarks to the Author:

General comments

The overall tone and structure of this manuscript downplays the uncertainties, limitations, and ethical issues associated with research into molecular explanations for missing heritability in humans. While it is mentioned at the end of the first paragraph that such work is contentious, the very next sentence and much of the rest of the manuscript reads as uncontested fact. A manuscript on this topic should adopt a tone consistent with the tentative nature of much of the research, and should at least summarize the attendant ethical issues. This area of science is not only contentious because of the scientific problems and challenges, but also because of the ethical, legal, and social implications of this work, including concerns about the suggestion of heritable,

biological imprints of structural inequities exacerbating those inequities and generating a new biological determinism, legal and moral responsibility, and stigma.

While the authors do include a section on challenges for the field, they do not specifically call out the many limitations of this work, though they do allude to some of them in other sections. For example, very little of the human data presented include any data on the F3. The challenges section should also include: conducting meaningful informed consent with individuals across multiple generations for a longitudinal study, exposures under study that call for mitigation, and the ethical and policy implications of the work. Finally, there is insufficient clarity in the early sections about when the authors are talking about data from humans and when they are talking about data from non-human animals.

Specific comments

P3¶5 In the first sentence, why are chromosomal aneuploidies highlighted first, rather than starting with SNPs?

P4¶4 What is the hypothesis for how such changes escape epigenetic reprogramming?

P5¶2 "...affect semen quality in adulthood and possibly the health of future generations."  this is a big, unsupported leap

¶4 Please define 'OS' at first use

The last sentence, "Future studies are needed..." suggests that none of the data presented in the previous three paragraphs demonstrates transgenerational epigenetic health effects, though that is not how the section is framed. This is the sort of disconnect in tone highlighted in the general comments above.

¶5 "...environmental exposures in the parent generation, such as TO particulate matter..."  the word 'to' appears to be missing in this sentence

P6¶2 "A few STUDIES included exposure during..."  Recommend adding 'studies' for clarity

P7¶1 "...normal expression of postsynaptic DENSITY genes during..."  'densities' in this sentence should be 'density'

"...transcriptomic changes in offspring and grand-offspring."  recommend changing to F1 and F2 for consistency, assuming these are rodent data

¶2 This paragraph is confusing. The first sentence talks about human studies of "intergenerational and transgenerational effects"  effects of what? In the next sentence, the mechanisms for these effects include not only epigenetic, but also genetic and social/behavioral explanations. This subtly alludes to the controversy surrounding this work without actually naming it or engaging with it.

¶3 In the title for this section, should the 'of' be 'on'?

The very first data/example highlighted could very plausibly have an entirely social/environmental explanation, including multiple generations of a family living in a zip code with poor air quality, housing stock, health care access, etc., but this is not addressed. Such possibilities are not mentioned until the last sentence in the section.

P8¶4 This paragraph sounds like a counter-argument to the epigenetic hypothesis

P9¶4 "...epigenetic, environmental AND SOCIAL factors on cross-generational..."  recommend

adding "and social" to this sentence, as indicated

¶5 "...meta-analysis of 13 non-clinical samples comprising 1,012 participants..."  it is unclear what this means. Do the authors mean 13 studies involving 1,012 Holocaust survivors? What is meant by 'non-clinical'?

Reviewer #3:

Remarks to the Author:

The review is well referenced and written, covers all the main topics of the research area and highlights well contradictions and research challenges.

Much of this has been repeated elsewhere but not previously in combination with a good description of the highly promising Environmental influences on Child Health Outcomes (ECHO) study. More is needed in this review that introduces the ECHO study. For example, how was it set up, where are the study populations located, when was it started, how will it differ from other large cohort studies e.g. ALSPAC?

The section on 'The Possibility for Transgenerational Epigenetic Health Effects' is inappropriately named as it contains very little on transgenerational effects (as defined by the authors i.e. an effect on a generation that did not directly experience the exposure). The section mainly describes direct effects of exposures on reproductive tissues which in the case of males are often genetic, not epigenetic.

On page 4 the references 20, 21, 25-39 are provided as evidence for intergenerational and transgenerational epigenetic modification by prenatal exposures. Please separate the references into those that show the former and those that support the latter - it seems that nearly all references only investigated intergenerationally inherited epigenetic states.

In the same paragraph the review is perpetuating a common oversimplification of the types of genes which are affected by prenatal exposures i.e. only reporting alteration of genomically imprinted genes. In fact the systematic review that is referenced (ref 47) contains many valuable examples of epigenetic changes at non-imprinted genes. The relatively high proportion of evidence for environmental alteration of imprinted genes was heavily influenced by the prior large body of literature on the powerful developmental roles that these genes have (along with their established intergenerational epigenetic inheritance). Because of this literature they were the first genes to be investigated, in a subjective manner, and continue to be a focus due to the perpetuation of their presumed primacy. However, subsequent unbiased studies have downplayed the role of these famous genes e.g. PLoS Genet. 2012;8(4):e1002605. More balance is needed in this section.

Understanding environmental influences on child health across generations

Response to reviewers' comments

Reviewer #1:

In this manuscript, Breton et al. compiled vast amounts of epidemiology literature to provide a comprehensive summarization of the inter- and trans-generational effects of environmental exposures on health outcomes in both animals and humans. Overall, this is a well-organized and insightful review. The authors covered a very diverse set of environmental factors and health outcomes, and considered both maternal and paternal exposures, thus providing an excellent resource for this complicated topic. The manuscript focused heavily on epidemiology literature that provides clinical observations to associate environmental exposures to inheritable health outcomes, yet lacking in mechanistic understanding of the process and its relation to epigenetics. However, these are known limitations and challenges in the field and are well-acknowledged by the authors.

The following suggestions may strengthen this manuscript:

1. In “Higher levels of concordance in genome-wide methylation patterns between monozygotic vs. dizygotic twins suggest an interplay between genetics and epigenetics.” (page 4), it would be more accurate to state that this high concordance of methylation patterns in monozygotic twins appears in the early years of life according to study by Fraga et. al. 2005. PNAS. PMID: 16009939.

Response. Thank you for your suggestion. We have edited the language to reflect the suggestion. (line 128-129, page 4).

Revised text now reads “Higher levels of concordance in genome-wide methylation profiles between monozygotic vs. dizygotic twins suggest an interplay between genetics and epigenetics in the early years of life.”

2. In “OS-mediated sperm DNA fragmentation and/or epigenetic changes may underlie observed associations of a wide variety of paternal environmental and lifestyle factors with birth defects and childhood diseases” on page 5, what does “OS-mediated” refer to? Maybe ROS-mediated? Besides, one study reported that impairments of spermatogenesis (e.g. decrease in sperm count and sperm motility) that induced by hypoxia exposure are associated with methylation patterns (Wang et. al. 2016. Nat. Commun. PMID: 27373813), which should be considered.

Response. We apologize for failing to define the abbreviation. We have edited this sentence to define OS as oxidative stress and we have included some additional sentences addressing the reviewer’s suggestion regarding hypoxia (line 189-195, page 5).

Revised text now reads “One of the most notable causes of sperm DNA fragmentation is oxidative stress (OS).^{72,73} Sperm are particularly vulnerable to genetic damage due to OS because sperm heads, which are filled with tightly-packed chromatin, lack cytoplasm that contains the enzymes necessary for DNA repair.⁷⁴ OS may also affect the sperm epigenome, as some studies have shown hypoxia to be associated both with impairments in spermatogenesis and alterations in DNA methylation.⁷⁵⁻⁸⁰ In general, it is theorized that OS-mediated sperm DNA fragmentation and/or epigenetic changes may underlie observed

associations of a wide variety of paternal environmental and lifestyle factors with birth defects and childhood diseases.^{81,82}

3. In the section “Inter- and transgenerational effects of birthweight”, the authors have reviewed the transgenerational influences of social, environmental, and metabolic factors on the birth weight. It might be necessary to discuss the potential epigenetic mechanisms that transmit these effects to birth weight of generations as reported in these studies, e.g. Küpers et. al. 2019. Nat. Commun. PMID: 31015461; Lin et. al. 2017. BMC med. PMID: 28264723; Agha et. al. 2016. Clin Epigenetics. PMID: 27891191.

Response. Thank you for this suggestion. The intergenerational effects are now discussed and the suggested papers as well as others have been cited (line 365-407, page 9).

4. Reference 17 does not seem to contain any information regarding epigenetics, might consider removing

Response. We have removed this reference.

5. Reference 98 does not seem to contain any information regarding to epigenetics, might consider removing

Response. We have removed this reference.

Reviewer #2:

General comments:

1. The overall tone and structure of this manuscript downplays the uncertainties, limitations, and ethical issues associated with research into molecular explanations for missing heritability in humans. While it is mentioned at the end of the first paragraph that such work is contentious, the very next sentence and much of the rest of the manuscript reads as uncontested fact.

Response. We thank the reviewer for his/her careful critique and concerns about ethical issues. We have endeavored to find a more balanced tone throughout the manuscript, beginning in the aforementioned paragraph, in order to more genuinely reflect the current lack of agreement in the field over evidence of transgenerational epigenetic transmission of environmental exposures. We have also added a section at the end (line 569-599, page 13) to more fully discuss the ethical considerations and concerns in epigenetic research. We welcome continued feedback on this important issue should the reviewer or editor feel that further revision is necessary.

2. A manuscript on this topic should adopt a tone consistent with the tentative nature of much of the research, and should at least summarize the attendant ethical issues. This area of science is not only contentious because of the scientific problems and challenges, but also because of the ethical, legal, and social implications of this work, including concerns about the suggestion of heritable, biological imprints of structural inequities exacerbating those inequities and generating a new biological determinism, legal and moral responsibility, and stigma. While the authors do include a section on challenges for the field, they do not specifically call out the many limitations of this work, though they do allude to some of them in other sections. For example, very little of the human data presented include any data on the F3. The challenges section should also include: conducting meaningful informed consent with individuals across

multiple generations for a longitudinal study, exposures under study that call for mitigation, and the ethical and policy implications of the work.

Response. We have now made reference to the ethical, legal and social implications in the introduction (line 79-80, page 3). We have also changed wording and tone throughout the manuscript and have added a section entitled “Ethical, Legal and Social Implications of Epigenetic Research” right before the Conclusion (line 569-599, page 13). For instance, on page 12, line 517-519, we have added the following sentence to address concerns about informed consent: “Moreover there may be ethical concerns about using samples from subjects whose informed consent was provided years or decades ago and may not directly address these types of assays in biospecimens.” We have also addressed the fact that very little human data on F3 exist, as is mentioned in the first paragraph under Current Operational and Analytic Challenges in the Field (line 496-500, page 11), which reads: “Inter- and transgenerational epidemiological studies face several obstacles to their success (Table 1, page 29). Perhaps the greatest operational challenge to performing intergenerational and transgenerational investigations in human populations is the lengthy time (20-40 years) between generations. As a result, there is a paucity of data on the F3 generation in humans.”

3. Finally, there is insufficient clarity in the early sections about when the authors are talking about data from humans and when they are talking about data from non-human animals.

Response. We have now clarified that most of the data presented are from human studies with the exception of the section titled “Animal Models of Epigenetic Inheritance” (line 226, page 6). If animal studies are alluded to outside of this section, we have made sure to describe them as such.

Specific comments:

1. P3¶5 In the first sentence, why are chromosomal aneuploidies highlighted first, rather than starting with SNPs?

Response. There was no particular reason for the order in which types of mechanisms were listed. We have reworded this sentence to decrease emphasis on any one type, by simply listing them all as follows. “Mechanisms of *genetic inheritance* include the following: germline chromosomal aneuploidies, germline DNA sequence variations such as single nucleotide polymorphisms (SNPs), small segments of nucleotide insertions and deletions, and larger structural variants, as well as tandem repeats and retrotransposons” (line 110, Page 4).

2. P4¶4 What is the hypothesis for how such changes escape epigenetic reprogramming?

Response. We have added a sentence here to help explain how changes might escape reprogramming (line 212-215, page 6). One hypothesized molecular mechanism for bypassing the reprogramming wave is through small regulatory RNAs, sequentially generated in parental somatic tissues, packaged in extracellular vesicles (ECVs), and delivered to early embryos, where they ultimately drive a global reprogramming of genome expression.

3. P5¶2 “...affect semen quality in adulthood and possibly the health of future generations.”  this is a big, unsupported leap

Response. We have removed this concluding sentence and replaced it with the following: "The consequences of such effects on the future progeny are largely unknown." (line 157, Page 5)

4. ¶4 Please define 'OS' at first use

Response. We have defined OS as oxidative stress (line 189, page 5).

5. The last sentence, "Future studies are needed..." suggests that none of the data presented in the previous three paragraphs demonstrates transgenerational epigenetic health effects, though that is not how the section is framed. This is the sort of disconnect in tone highlighted in the general comments above.

Response. Thank you for pointing out this discrepancy. We have reworded and reorganized the placement of this entire section to reflect environmental, rather than transgenerational, effects on the germline to be more consistent with the cited literature and to balance tone of the article (line 145, Page 4 section titled "Germline Changes Associated with Environmental Exposures".) Evidence for transgenerational effects comes primarily from animal studies, which are presented later in the manuscript. However, presenting evidence of environmental effects on the germline is the first step toward understanding the possibility of a mechanism for transgenerational effects in humans.

6. ¶5 "...environmental exposures in the parent generation, such as TO particulate matter..."  the word 'to' appears to be missing in this sentence

Response. We have made the suggested revision (line 229-230, page 6).

7. P6¶2 "A few STUDIES included exposure during..."  Recommend adding 'studies' for clarity

Response. We have made the suggested revision (line 245, page 6).

8. P7¶1 "...normal expression of postsynaptic DENSITY genes during..."  'densities' in this sentence should be 'density'

Response. We have made the suggested revision (line 304, page 7).

9. "...transcriptomic changes in offspring and grand-offspring."  recommend changing to F1 and F2 for consistency, assuming these are rodent data

Response. Thank you. We have made the suggested changes (line 306, page 7).

10. ¶2 This paragraph is confusing. The first sentence talks about human studies of "intergenerational and transgenerational effects"  effects of what? In the next sentence, the mechanisms for these effects include not only epigenetic, but also genetic and social/behavioral explanations. This subtly alludes to the controversy surrounding this work without actually naming it or engaging with it.

Response. Thank you for suggesting the need for further clarification. We have rephrased this introductory paragraph as follows to provide clarity and set the stage for more discussion of the

debatable hypotheses in the ensuing paragraphs and discussion sections. “Human studies that document both intergenerational and transgenerational effects of environmental exposures on health outcomes are relatively rare. This section presents available evidence, covering a range of environmental exposures and health outcomes. The underlying mechanisms that might explain these effects have been neither uniformly nor comprehensively tested and are, in fact, the subject of some debate. While the focus of the current manuscript is on hypothesized epigenetic mechanisms of intergenerational and transgenerational transmission of environmental exposure effects, we acknowledge and discuss the challenges in disentangling potential epigenetic mechanisms from concurrent genetic, ecological and sociocultural factors that may track across generations.” (line 309-316, page 8).

11. ¶3 *In the title for this section, should the ‘of’ be ‘on’?*

Response. We have made this suggested change to all sub headers (line 321, page 8).

12. *The very first data/example highlighted could very plausibly have an entirely social/environmental explanation, including multiple generations of a family living in a zip code with poor air quality, housing stock, health care access, etc., but this is not addressed. Such possibilities are not mentioned until the last sentence in the section.*

Response. We have rewritten this introductory paragraph to more explicitly call out and differentiate between genetic, epigenetic, socio-cultural and environmental explanations for asthma risk.(line 327-331, Page 8) Specifically, we note that “non-biological explanations and interactions between genetic and non-genetic risk factors are also feasible. For example, multiple generations of families living in low-socioeconomic neighborhoods with higher exposure burden and greater poverty may also explain or exacerbate asthma risk.”

13. P8¶4 *This paragraph sounds like a counter-argument to the epigenetic hypothesis*

Response. The section on “Evidence in Humans” (beginning on line 421) is intended to show data from the literature relating an exposure in the grandparental (or great-parental) generation on a health outcome (as indicated by the sub header). In some cases, there may be evidence to suggest epigenetic or other biologic mechanisms underlie these effects, and in others there may be more evidence to support socio-cultural, demographic mechanisms. In addition, in some cases, evidence may point to purely environmental (chemical) influences, whereas in others there may be no additional evidence for any mechanism driving the observed associations. In the case of birthweight, we include a paragraph discussing social influences and a second paragraph discussing environmental influences such as smoking and health characteristics such as BMI. In both paragraphs we now note the caveat that these studies have largely NOT addressed the potential for epigenetic mechanisms to play a role and that there is a need to do so. (beginning on line 503 and again on line 527)

14. P9¶4 *“...epigenetic, environmental AND SOCIAL factors on cross-generational...”  recommend adding “and social” to this sentence, as indicated*

Response. We have made this suggested improvement (line 461, page 10).

15. ¶5 *“...meta-analysis of 13 non-clinical samples comprising 1,012 participants...”  it is unclear what this means. Do the authors mean 13 studies involving 1,012 Holocaust survivors? What is meant by ‘non-clinical’?*

Response. We have clarified this text as follows: “In a meta-analysis of 13 community samples representing 1,012 children with or without a grandparent who was a Holocaust survivor, Sagi-Schwartz and colleagues concluded that there was no evidence for tertiary traumatization.²¹⁹” The term non-clinical was used to indicate that the samples were not identified in clinical settings. (line 465-469, Page 11)

Reviewer #3:

1. The review is well referenced and written, covers all the main topics of the research area and highlights well contradictions and research challenges. Much of this has been repeated elsewhere but not previously in combination with a good description of the highly promising Environmental influences on Child Health Outcomes (ECHO) study. More is needed in this review that introduces the ECHO study. For example, how was it set up, where are the study populations located, when was it started, how will it differ from other large cohort studies e.g. ALSPAC?

Response. Thank you for this helpful comment. We have now included a paragraph describing ECHO more in depth under the header “An opportunity: The ECHO Study” starting on line 601, page 13.

2. The section on 'The Possibility for Transgenerational Epigenetic Health Effects' is inappropriately named as it contains very little on transgenerational effects (as defined by the authors i.e. an effect on a generation that did not directly experience the exposure). The section mainly describes direct effects of exposures on reproductive tissues which in the case of males are often genetic, not epigenetic.

Response. Thank you for pointing this out. We have renamed this section “Germline Changes Associated with Environmental Exposures” (line 145, Page 4). We also reordered the sections and moved this section up to follow Genetic and Epigenetic Mechanisms.

3. On page 4 the references 20, 21, 25-39 are provided as evidence for intergenerational and transgenerational epigenetic modification by prenatal exposures. Please separate the references into those that show the former and those that support the latter - it seems that nearly all references only investigated intergenerationally inherited epigenetic states.

Response. Thank you for this attention to detail. In fact, references 20 and 21 (now 80 and 81) refer to biological mechanisms generally and do not address exposure effects. The other references (now 85-99) all support intergenerational effects. Therefore, we have amended the sentence to read: “Epigenetic modifications may be induced by prenatal exposures and can be inherited intergenerationally, escaping the major waves of epigenetic reprogramming that occur during fertilization and gametogenesis.” (line 210, page 6)

4. In the same paragraph the review is perpetuating a common oversimplification of the types of genes which are affected by prenatal exposures i.e. only reporting alteration of genomically imprinted genes. In fact the systematic review that is referenced (ref 47) contains many valuable examples of epigenetic changes at non-imprinted genes. The relatively high proportion of evidence for environmental alteration of imprinted genes was heavily influenced by the prior large body of literature on the powerful developmental roles that these genes have (along with their established intergenerational epigenetic inheritance). Because of this literature they were the first genes to be investigated, in a subjective manner, and continue to be a focus due to the

perpetuation of their presumed primacy. However, subsequent unbiased studies have downplayed the role of these famous genes e.g. PLoS Genet. 2012;8(4):e1002605. More balance is needed in this section.

Response. We thank the reviewers for pointing out this imbalance. In an effort to be concise and to provide greater balance, we removed the sentences that provided examples specific to effects on imprinted genes only. Instead, we now discuss these effects more generally without placing undue influence on imprinted genes. (beginning on line 295)

Reviewers' Comments:

Reviewer #1:

Remarks to the Author:

The authors have addressed my concerns. This is an interesting study.

Reviewer #3:

Remarks to the Author:

I thank the authors for their consideration of my previous comments.

I only have 2 new minor points.

The sentence in lines 566-568 seems to be missing a word or two. What "very significantly"?

With regard to the sentences starting on line 194 that describes single-generation epigenetic effects, I think that it is important to include a comment that these instances avoid the major reprogramming events (which are mentioned at the start of the paragraph) by occurring after fertilisation and due to direct exposure to the developing somatic tissues.

Reviewer #4:

Remarks to the Author:

Thank you for submitting a pretty thorough revision of the manuscript. While I found that the authors have addressed the majority of the original comments and suggestions, there are some thoughts and further suggestions to strengthen the manuscript.

1. Specific comment #2: I think that "escape" of epigenetic reprogramming during early embryonic development needs to be distinguished from "epigenetic reprogramming" or "developmental reprogramming". I believe that the specific mechanism presented by the authors comes from Spadafora review (Environmental Epigenetics) and it is specific to the potential role of RNA-containing ECVs in reprogramming expression profiles in the embryos (this would be a reprogramming mechanism). "Escape" of epigenetic reprogramming would be best illustrated by active mechanisms that act of repetitive elements and imprinted genes during early embryonic development (in which the failure to do so also leads to reprogramming of development).
2. The stated line #, Page # in the rebuttal did not match the final revised manuscript. It was difficult to find what was corrected, and I am sorry but I was not able to review Specific Comment #3.
3. In response to Specific comment #13: There are some DNA methylation analysis that has been done in human studies that would fall in the categories of "intergenerational" populations in this review. For example the IGF2 methylation in Dutch Famine Cohort, NR3C1 methylation in the Tutsi genocide cohorts, and FKBP5 methylation in Holocaust survivors. I think that including these information will strengthen the review.

Exploring the evidence for epigenetic regulation of environmental influences on child health across generations

Response to Reviewers comments May 8th, 2021

REVIEWERS' COMMENTS:

Reviewer #3 (Remarks to the Author):

I thank the authors for their consideration of my previous comments.

I only have 2 new minor points.

The sentence in lines 566-568 seems to be missing a word or two. What "very significantly"?

Response: We apologize for the typo here. The correct phrasing was "vary significantly." i.e. "...extant cohorts that are a part of the ECHO program vary significantly in their study design, year of initiation...." (line 583, page 13).

With regard to the sentences starting on line 194 that describes single-generation epigenetic effects, I think that it is important to include a comment that these instances avoid the major reprogramming events (which are mentioned at the start of the paragraph) by occurring after fertilization and due to direct exposure to the developing somatic tissues.

Response: Thank you for this point. We have added a sentence to include this comment. (line 197, page 5).

Reviewer #4 (Remarks to the Author):

Thank you for submitting a pretty thorough revision of the manuscript. While I found that the authors have addressed the majority of the original comments and suggestions, there are some thoughts and further suggestions to strengthen the manuscript.

1. Specific comment #2: I think that "escape" of epigenetic reprogramming during early embryonic development needs to be distinguished from "epigenetic reprogramming" or "developmental reprogramming". I believe that the specific mechanism presented by the authors comes from Spadafora review (Environmental Epigenetics) and it is specific to the potential role of RNA-containing ECVs in reprogramming expression profiles in the embryos (this would be a reprogramming mechanism). "Escape" of epigenetic reprogramming would be best illustrated by active mechanisms that act of repetitive elements and imprinted genes during early embryonic development (in which the failure to do so also leads to reprogramming of development).

Response: Thank you for your suggestion. We have clarified the language in this section and added a new sentence addressing additional evidence for "escapees". The section now reads "One hypothesized molecular mechanism for bypassing the DNA-methylation reprogramming wave is through small regulatory RNAs, sequentially generated in parental somatic tissues, packaged in extracellular vesicles (ECVs), and delivered to early embryos, where they ultimately drive a global reprogramming of genome expression. Other means of escaping the early embryonic reprogramming are evidenced by CpG loci adjacent to intracisternal-A-particle. elements or telomeric regions. Single-generation epigenetic effects may also

occur when exposures directly affect the developing somatic tissue post-fertilization.” (starting on line 192, page 5).

2. The stated line #, Page # in the rebuttal did not match the final revised manuscript. It was difficult to find what was corrected, and I am sorry but I was not able to review Specific Comment #3.

Response: We apologize for that error.

3. In response to Specific comment #13: There are some DNA methylation analysis that has been done in human studies that would fall in the categories of “intergenerational” populations in this review. For example the IGF2 methylation in Dutch Famine Cohort, NR3C1 methylation in the Tutsi genocide cohorts, and FKBP5 methylation in Holocaust survivors. I think that including these information will strengthen the review.

Response: Thank you for the additional suggestions. We have added the evidence related to trauma in a paragraph in the section under neurodevelopmental outcomes (starting on line 465, page 11) and evidence related to the Dutch famine in the section on cardiovascular health (line 397, page 9).